# Mathematical Modelling Using Predictive Biomarkers for the Outcome of Canine Leishmaniasis upon Chemotherapy

**DOI:** 10.3390/microorganisms8050745

**Published:** 2020-05-15

**Authors:** Rafaela de Sousa Gonçalves, Flaviane Alves de Pinho, Ricardo Jorge Dinis-Oliveira, Rui Azevedo, Joana Gaifem, Daniela Farias Larangeira, Eduardo Milton Ramos-Sanchez, Hiro Goto, Ricardo Silvestre, Stella Maria Barrouin-Melo

**Affiliations:** 1Laboratory of Veterinary Infectious Diseases, Teaching Hospital of Veterinary Medicine, Federal University of Bahia, Salvador 40170-110, BA, Brazil; rdsgon@gmail.com (R.d.S.G.); flaviane.alves@ufba.br (F.A.d.P.); daniela.larangeira@ufba.br (D.F.L.); 2Life and Health Sciences Research Institute (ICVS), School of Medicine, University of Minho, 4710-057 Braga, Portugal; joana.f.gaifem@gmail.com; 3ICVS/3B’s-PT Government Associate Laboratory, 4710-057 Braga/Guimarães, Portugal; 4Department of Veterinary Anatomy, Pathology and Clinics, School of Veterinary Medicine and Zootechny, Federal University of Bahia, Salvador 40170-110, BA, Brazil; 5Department of Public Health and Forensic Sciences, and Medical Education, Faculty of Medicine, University of Porto, 4200-319 Porto, Portugal; ricardinis@sapo.pt; 6Department of Sciences, IINFACTS—Institute of Research and Advanced Training in Health Sciences and Technologies, University Institute of Health Sciences (IUCS), CESPU, CRL, 4585-116 Gandra, Portugal; rui.azevedo@iucs.cespu.pt; 7UCIBIO-REQUIMTE, Laboratory of Toxicology, Department of Biological Sciences, Faculty of Pharmacy, University of Porto, 4050-313 Porto, Portugal (R.J.D.-O.); 8Laboratory of Seroepidemiology and Immunobiology, Institute of Tropical Medicine, Faculdade de Medicina, Universidade de São Paulo, Sao Paulo 05403-000, SP, Brazil; eduardors22@hotmail.com (E.M.R.-S.); hgoto@usp.br (H.G.); 9Departamento de Salud Publica, Facultad de Ciencias de La Salud, Universidad Nacional Toribio Rodriguez de Mendoza de Amazonas, Chachapoyas 01000, Peru; 10Department of Preventive Medicine, Faculdade de Medicina, Universidade de São Paulo, 05403-000 São Paulo, SP, Brazil

**Keywords:** mathematical model, treatment, hematological parameters, biochemical parameters, *Leishmania*

## Abstract

Prediction parameters of possible outcomes of canine leishmaniasis (CanL) therapy might help with therapeutic decisions and animal health care. Here, we aimed to develop a diagnostic method with predictive value by analyzing two groups of dogs with CanL, those that exhibited a decrease in parasite load upon antiparasitic treatment (group: responders) and those that maintained high parasite load despite the treatment (group: non-responders). The parameters analyzed were parasitic load determined by q-PCR, hemogram, serum biochemistry and immune system-related gene expression signature. A mathematical model was applied to the analysis of these parameters to predict how efficient their response to therapy would be. Responder dogs restored hematological and biochemical parameters to the reference values and exhibited a Th1 cell activation profile with a linear tendency to reach mild clinical alteration stages. Differently, non-responders developed a mixed Th1/Th2 response and exhibited markers of liver and kidney injury. Erythrocyte counts and serum phosphorus were identified as predictive markers of therapeutic response at an early period of assessment of CanL. The results presented in this study are highly encouraging and may represent a new paradigm for future assistance to clinicians to interfere precociously in the therapeutic approach, with a more precise definition in the patient’s prognosis.

## 1. Introduction

Canine leishmaniasis (CanL) is a severe disease caused by the protozoa Leishmania infantum [1,2,3]. As a zoonotic infection, L. infantum is a major health concern, since it also causes human visceral leishmaniasis, a neglected tropical disease, in the Americas, Europe, North Africa, the Middle East and China [4]. Dogs are susceptible to the development of high parasitic loads, playing a key role in the transmission cycle of L. infantum via phlebotomine sand flies in endemic areas [5,6]. Higher parasite loads in dogs are related to more severe clinical disease [7] and higher infectiousness to sand flies [5]. As full-blown disease is associated with active infection [8], efficacy of CanL treatment should be primarily centered in its capability of reducing L. infantum load.

Treatment of CanL is a challenge for veterinary practitioners regarding anti-parasitical efficacy, adverse effects and cost. Current leishmanicidal drugs, such as antimonials, miltefosine and allopurinol, can clinically cure dogs or temporarily improve clinical signs, but no evidence of efficacy in terms of a parasitological cure exists for any of them [8,9]. Consequently, clinical recurrences associated with high parasite loads in a number of treated dogs are frequent at diverse time intervals post treatment [10]. In Brazil, where most of the South American cases of *L. infantum* infection occur, the use of antimonials for canine treatment is prohibited [11].

The success of some antileishmanial protocols in reducing *L. infantum* load and blocking parasite transmission to phlebotomine vectors have been evidenced by xenodiagnoses [12,13,14]. However, failure of anti-*Leishmania* therapy depends on drug pharmacodynamic effects, parasite’s inherent virulence or drug resistance development and host’s immune and nutritional status, or co-infections, which contribute decisively to the recurrence [15,16]. Prognosis of response to treatment largely depends on the severity of the visceral disease, as determined by the clinicopathological alterations, particularly those related to renal function [17,18].

Because dogs respond differently to anti-parasite treatments and there are still no predictive tools to discriminate dogs that respond to therapy from those that maintain high parasitic loads despite treatment, we aimed at devoting the present study to this aspect. This study was a retrospective longitudinal analysis of parasitological, biochemical and hematological parameters in a cohort of dogs treated for CanL with second-line antileishmanial drugs, as reported in the literature [8,9,19,20,21,22]. The study’s objective was to identify biomarkers and to develop a prototype mathematical model with predictive value to estimate the success of a leishmanicidal therapy as a tool for improved diagnosis and treatment monitoring.

## 2. Materials and Methods 

### 2.1. Ethics Approval

This work was approved by the Ethics Committee on Animal Use of the Federal University of Bahia, Brazil (CEUA-UFBA, n. 19/2011), substantiated in the bioethical principles of animal experimentation. All procedures were followed in accordance with the guidelines of the Brazilian Council of Animal Experimentation (CONCEA—Conselho Nacional de Controle de Experimentação Animal) and strictly followed the Brazilian law for “Procedures for the Scientific Use of Animals” (11.794/2008). All manipulation in dogs, whether invasive or non-invasive, was performed or monitored by veterinarians. The group of non-responder dogs was designed based on the parasitic load whose quantitative test was performed at the end of the study. Therefore, throughout the course of the study, dogs that developed other diseases or acquired co-infections were removed to receive proper additional treatment. 

### 2.2. Animals and Sampling

A one-year longitudinal analysis of hematological and biochemical parameters was conducted in 26 dogs, naturally infected by *L. infantum* in the state of Bahia in Brazil, submitted to leishmanicidal treatment. The inclusion criteria were confirmed infection with *Leishmania* spp. via parasitological and/or molecular techniques, and consent of the guardians for the treatment with the multi-drug antiparasitic regimen (metronidazole + ketoconazole + allopurinol). The coexistence of other diseases or co-infections and terminal CanL were the exclusion criteria in the beginning and at any other time of examination in the course of the follow-up. Dogs were evaluated at six different time points during the year. The time 0 (T0) consisted of the moment in which the dogs arrived at the Zoonotic Leishmaniasis Outpatient Clinic at the Teaching Hospital of Veterinary Medicine of the Federal University of Bahia (ALZ-UFBA) for diagnosis and first clinical care. Dogs were evaluated monthly during the first three months (T1, T2, T3), and then at the sixth (T6) and twelfth month (T12) post-treatment. At each evaluation, blood was collected from the jugular or cephalic vein and stored in EDTA tubes (BD Vacutainer; Becton, Dickinson) for hemogram and search for hematozoa. Blood was also collected into EDTA-free tubes (BD Vacutainer; Becton, Dickinson) for the determination of serological and biochemical parameters. Urine was collected by cystocentesis (in females) and urethral catheterization (in males) for urinalysis and urinary protein-creatinine ratio (UPC). For the collection of bone marrow and spleen samples by aspiration biopsy (performed at T0, T6 and T12), 20 mL syringes and needles 40 × 12 mm and 40 × 16 mm were used respectively, inserted in an anatomic-topographic region of the spleen [23] and sternum [24], with previous local antisepsis and mild sedation with acepromazine (0.02 mg/kg).

### 2.3. Clinical Evaluation and Staging

Dogs were clinically evaluated for the presence or absence of CanL-compatible clinical signs and definition of clinical staging, as determined by the LeishVet Group guidelines [8]. Weight loss, appetite alterations, facies, mucosal staining, dermatopathies (ulcers, cutaneous vasculitis, onychogryphosis, desquamation, alopecia, crusts, hyperkeratosis, depigmentation of snout, nodule), lymphadenopathy, ophthalmopathy, presence or absence of fever, diarrhea, epistaxis and enlargement of the spleen by palpation were assessed. Based on clinical and laboratory parameters, dogs were categorized in clinical staging I (mild disease), II (moderate disease), III (severe disease) or IV (very severe to terminal disease).

### 2.4. Anti-Leishmania Chemotherapy

Antiparasitic treatment consisted in the administration of metronidazole (25 mg/kg/twice daily) for 30 days associated to ketoconazole (10 mg/kg/once a day) for 40 days, followed by a maintenance treatment with allopurinol (10 mg/kg/twice daily) during at least one year. Prednisolone was given (beginning with 0.5 mg/kg/twice daily) for 30 days (including gradual withdrawal) [20,25,26] to minimize systemic immune complex formation, thus indirectly to prevent kidney injury during the four-week initial course of treatment with anti-leishmanial drugs. The multi-drug therapeutic regimen for CanL used in the present study has been previously described [14].

### 2.5. Hematological and Biochemical Parameters

Hemograms, characterization of blood cell morphology and search for hematozoa infection were performed as described [27]. The following biochemical parameters were measured: enzymes (alanine aminotransferase (ALT), aspartate transaminase (AST), creatine kinase (CK) and its isoenzyme creatine kinase MB (CK-MB), pseudocholinesterase and gamma-glutamyl transferase (GGT) (Cormay, Łomianki, Poland)), protein (albumin and total protein (Cormay, Łomianki, Poland)), substrates (creatinine, urea) and electrolytes (calcium, phosphorus and magnesium (Spinreact, Barcelona, Spain)). The urinary protein-creatinine (UPC) and serum calcium/phosphorus ratio were also quantified. All variables were measured in the automatic (Prestige 24i, Tokyo Japan) and semi-automatic (BioPlus-200, São Paulo Brazil) biochemical analyzer, as described [28].

### 2.6. Quantification of Total Globulin and Specific Anti-Leishmania Antibodies

For the quantification of total globulins, total protein values were subtracted from albumin values. Specific anti-*Leishmania* IgG antibodies were quantified in the sera as previously described [29], with minor modifications. Ninety-six-well plates (BioLegend, San Diego USA) were coated with 10 μg/mL of *L. infantum* soluble promastigote antigens. After washing with 3% PBS-low-fat-milk and 0.05% PBS-Tween (PBS-T), 100 μL/well of canine serum diluted 1:50,000 in PBS-T was added in duplicate and incubated at 30 min at 37 °C. The reaction was completed by adding the anti-dog IgG-HRP diluted at 1:5000 (Bethyl Laboratories, Montgomery, USA) for 30 min at 37 °C, and further incubation with 0.5 mg/mL of o-phenylenediamine dihydrochloride (OPD, Sigma, Darmstadt Germany) for 10 min. The reaction was stopped using 50 μL/well of 3 M HCl. Plates were read at 492 nm (Synergy, Biotek, Tokyo Japan).

### 2.7. Parasitic Load 

DNA of the collected aspirated bone marrow and spleen at T0, T6 and T12 was extracted using the commercial kit PureLink Genomic DNA^®^ (Invitrogen, Carlsbad USA) following the manufacturer’s recommendations. The quality and concentration of the DNA in each eluate were evaluated in the L-QUANT spectrophotometer (Loccus, São Paulo Brazil). For the quantification of *L. infantum* DNA in the spleen and bone marrow samples, qPCR assays were performed using the protocol described by Rolão et al. [30], with some modifications. The primers 5′-GGTTAGCCGATGGTGGTCTT-3′ (forward), 5′-GCTATATCATATGTCCAAGCACTTACCT3′ (reverse) and the probe TaqMan^®^ (Applied Biosystems Foster City USA) (5′- ACCACCTAAGGTCAACCC-3′) were used. The final reaction volume (25 μL) consisted of 4 μL of DNA, standardized at the concentration of 20 ng/μL, 12.5 mL of Mastermix Universal^®^ (Life Technology Corporation, Carlsbad USA), 5 μM of each primer and 10 μM of the probe TaqMan. qPCR was run at 95 °C for 10 min, 40 cycles at 95° C for 15 s and 60 °C for 1 min. All samples were analyzed in duplicate. Reactions were performed on the Applied Biosystems 7500 Real-Time PCR System (Life Technology Corporation Carlsbad USA). 

### 2.8. Quantitative PCR

Splenic mRNA expression of cytokines *IFN-γ, IL-2, TNF-α, IL-17A, IL-22, IL-4, IL-5* and *IL-10* was determined using real time PCR. Spleen fragments were macerated for total RNA extraction using Trizol reagent (Invitrogen, Carlsbad USA), as recommended by the manufacturer. RNA concentration was determined by OD260 measurement using a L-quant 1.0 spectrophotometer (Loccus^®^, São Paulo, Brazil). cDNA synthesis was performed using 1 µg of total RNA plus 10 µL of High-Capacity cDNA Reverse Transcription master mix (Applied Biosystems, Foster City USA), according to the manufacturer instructions. Real-Time quantitative PCR (qRT-PCR) reactions were run for each sample on a Bio-Rad CFX96 Real-Time System C1000 Thermal Cycler (Bio-rad Berkeley USA). Primer sequences were obtained from IDT (Lovaina, Belgium) and thoroughly tested. Specific oligonucleotides are: *IL2* (forward) CCCAAGAAGGCCACAGAATTTA, (reverse) TCCTTGGTGTCTGTCAAGTGAA; *IL4* (forward) CTAGCACTCACCAGCACCTT, (reverse) CACGAGTCGTTTCTCGCTGT; *IL5* (forward) GGCGATGGGAACCTGATGAT, (reverse) CGTGGGCAGTTTGGTTCTTC; *IL10* (forward) CAAGCCCTGTCGGAGATGAT, (reverse) AGAAATCGGTGACAGCGTCG; *IFNG* (forward) TCAAATTCCTGTGAACGATCTGC, (reverse) TTATTTCGATGCTCTGCGGC; *TNFA* (forward) CTCCAACTAATCAGCCCTCTTG, (reverse) GGGTTTGCTACAACATGAGCTACT; *GAPDH* (forward) GCTGGTGCTGAGTATGTTGTGGAG, (reverse) CAGCAGAAGGAGCAGAGATGATGA. The RT product was expanded using the NZYSpeedy qPCR Green Master Mix kit (NZYTech, Lisbon Portugal) and results were normalized to the expression of the housekeeping gene Gapdh. After amplification, cycle threshold-values (Ct-values) were calculated for all samples and gene expression changes were analyzed in the CFX Manager Software (Bio-Rad) and represented as arbitrary units (AU).

### 2.9. Statistical Analysis

The Kruskal–Wallis test followed by the Dunn test were used for data analysis between two groups. A one-way analysis of variance (ANOVA) followed by a Bonferroni’s post-hoc test was employed for multiple group comparisons. Exploratory logistic regression models were made in R using the package Bias Reduction in Binomial-Response Generalized Linear Models [31,32]. The Chi-square test was used for statistical analysis to evaluate the significant association between variables. Data are reported as means ± standard deviation (SD). Statistically significant values are as follows: * *p* < 0.05, ** *p* < 0.01, *** *p* < 0.001. 

## 3. Results

### 3.1. Responder Dogs Exhibit a Decrease in Splenic and Bone Marrow Parasite Loads and Improvement of Clinical Staging

Leishmania infantum absolute parasite load evolution in spleen and bone marrow of a cohort of dogs under leishmanicidal treatment for 12 months were distinct between two groups. Dogs that presented a significant reduction in parasite loads from T0 to T12 were grouped as chemotherapy Responders (GR, 46%, 12 dogs), while those that failed to have the parasite burden reduced were grouped as Non-Responders (GNR, 54%, 14 dogs; Figure 1A, B). At T0, before treatment, there were no parasitic loads’ differences between groups in both organs (Figure 1A, B), as well as in the clinical staging (Figure 1C). At T6, GR exhibited a noticeable parasitic load reduction in comparison to that of GNR. After one year of treatment (T12), GR’s parasitic loads were significantly lower than those at T0 (*p* < 0.001) in both tissues (Figure 1A, B), having decreased 97.3% and 98.5% in bone marrow and spleen, respectively. In opposition, no significant difference was observed for GNR (Figure 1A, B). At T0, both groups presented the same clinical stage—II, characterized by moderate disease. Accompanying the parasite load data, at T12, GR exhibited on average a mild disease (stage I), while GNR remained in stage II (Figure 1C). 

### 3.2. Responder Dogs Display a Th1 Signature

We next addressed the immune response developed by GR and GNR. The median monocytes’ counts was similar and within the reference range in both groups (Figure 2A). The lymphocytes’ counts, however, changed at T3 and T6 (Figure 2B), not only increasing from T2 to T3 within GR, but also becoming consistently higher than those of GNR at T3 and T6. To further explore, we performed a qPCR analysis on spleen samples at T0, T6 and T12. The GR exhibited a Th1 signature, with increased levels of *IL2*, *IFNG* and *TNFA* transcripts at T6 (Figure 2C), while no differences were found in *IL17* nor *IL22* (Appendix A). Moreover, a significant reduction on *IL10* transcripts of GR was accompanied by similar levels of Th2-associated *IL4* and *IL5* cytokines (Figure 2C), indicating an effective Th1 response underlying the protective response. Remarkably, these effects on transcripts could be verified in all individual dogs analyzed (Appendix A). Overall, our data showed that increased levels of blood lymphocytes and a splenic Th1 response guide the response to therapy.

### 3.3. Non-Responder Dogs Failed to Restore the Hematological Parameters to Normal Reference Values

At T0, average GR dogs displayed median erythrocyte counts, hemoglobin and hematocrit levels within the normal reference range, while in GNR dogs, those parameters were sub-optimal (Figure 3A–C). Of note, at T0, 8 out of 14 (57.14%) of GNR had anemia; of these, 4 out of 8 (50%) presented a reticulocyte counting below 60 × 10^3^/mm^3^ (had non-regenerative anemia). Regarding the GR dogs, 3 out of 12 (25%) were anemic, with 2 of these presenting a reticulocyte counting superior to 60 × 10^3^/mm^3^ (had regenerative anemia). At T12, 9 out of 13 (69.2%) GNR dogs were anemic, compared to none of the GR dogs; among those 9 dogs, only one presented a reticulocyte counting superior to 60 × 10^3^/mm^3^. This is strongly suggestive that non-regenerative anemia indeed progressed in the GNR group of dogs during the one year of follow up, despite treatment. Total globulin levels were increased in both groups, but the GR displayed a significantly lower level (Figure 3D). Interestingly, the difference in globulins was not equivalent to anti-Leishmania-specific IgG titters, which could not discriminate GR from GNR (Figure 3E). 

The cohort of dogs was monitored during one year through hematological and biochemical analysis performed at one-, two-, three-, six- and twelve-months post-treatment (T1, T2, T3, T6 and T12) (Figure 3A–D). Although a tendency for hematological improvement was observed in all parameters until the third month of treatment, non-responder dogs declined in months 6 and 12 (Figure 3A–D). In opposition, all responder dogs displayed erythrocyte numbers, hemoglobin and hematocrit within the normal reference values upon 12 months of treatment, apart from total globulins, where 3 out of 11 dogs still presented values above the upper limit (Figure 3A–D). Moreover, the observed recovery of responder dogs was accompanied by a significant reduction in the anti-Leishmania-specific IgG (Figure 3E). 

Finally, given that our data was suggestive that erythrocyte number, hemoglobin and hematocrit values can be predictive of the therapeutic response in a first moment of evaluation of sick dogs, before treatment, we calculated the relative risk (RR) and the 95% confidence interval (CI) to determine the likelihood of each of these parameters to be associated with the progression of CanL upon treatment. A significant association between decreased hemoglobin (3.14 RR with 1.14–8.7 CI), hematocrit (3.75 RR with 1.05–13.38 CI) and erythrocyte levels (1.82 RR with 0.88–3.73 CI) with the failure of CanL treatment was observed, reinforcing their utility to the prediction of therapeutic response. 

### 3.4. Serum Biochemistry Parameters Are Predictive of Visceral Organ Injury in Non-Responder Dogs

Serum levels of hepatocellular damage markers, alanine and aspartate transaminase (Appendix A), and biliary tract disease marker γ-glutamiltransferase (Appendix A), were within the reference range during follow-up in both groups. However, cholinesterase and albumin, proteins synthesized in the liver, were significantly reduced in the GNR at T0, T6 and T12 (Figure 4A and Appendix A). In GNR, while urea and creatinine values were within the reference range (Appendix A), the urinary protein/creatinine ratio (UPC), a predictor of renal disease, was elevated (Figure 4B). These peculiar changes are indicative of early hepatic and renal disease in the GNR. The values of hematological, serum biochemistry and parasitic load of the initial and final times of the responder and non-responder dogs to the treatment are shown in Table 1.

### 3.5. Development of a Mathematical Model with Predictive Value for the Success of CanL Chemotherapy

Apart from erythrocytes, hemoglobin and hematocrit (Figure 3A–C), phosphorous, and to a lesser extent, potassium and albumin also significantly discriminated GR from GNR at T0 (Figure 5A). High Spearman correlation coefficients were found between erythrocytes, hematocrit and hemoglobin (Figure 5B). Logistic regression models with the lowest Akaike information criterion (AIC) and higher accuracy using a leave-one-out cross-validation were selected for phosphorus plus one of the three highly correlated variables erythrocytes, hematocrit or hemoglobin (Figure 5C). The highest accuracy was obtained considering phosphorus and erythrocytes. For this case, the log(odds) of a successful treatment, using the full dataset, can be given by:log(odds) = 14.067 − 5.289 × [Phosphorus] + 1.170 × Number of erythrocytes(1)

The probability is obtained by applying the logistic function: (exp(x)/(1 + exp(x)), where x is the log(odds). Remarkably, this formula allowed 100% discrimination between GR and GNR, demonstrating the high predictive power of our mathematical model.

## 4. Discussion

Until recently, most studies of prognostic factors for treatment success or failure in leishmaniasis were restricted to humans [33,34,35,36,37], and the infected dog has been seen only as a protozoan reservoir. This is an innovative longitudinal study of prognostic factors in *L. infantum* naturally infected dogs under treatment that resulted in a clear distinction of two profiles of CanL dogs that responded differently to the same antiparasitic therapy, based on their tissue parasite load reduction. Dogs with reduced parasite numbers are less likely to develop clinical signs and are less infectious to vectors [5]. 

Studies indicate that the successful resolution of *Leishmania* infections depend on a response characterized by IFN-γ, IL-2 and TNF-α predominance, which increase phagocytic efficiency and lymphocyte cytotoxicity, triggering a protective immune response [38,39]. Accordingly, we observed that responder dogs exhibited a consistent Th1 signature that accompanied parasitism reduction upon treatment, in opposition to the non-responder group. 

In the present study, the standard parameters for clinical staging of CanL [8] and follow-up of the dogs, such as albumin/globulin ratio, non-regenerative anemia or renal functions, were assessed individually from the beginning to the end of the study. However, only at the end of the one-year study did the treatment outcome allow the determination of which dogs could be classified as responders or non-responders. The evident dichotomy between GR and GNR dogs allowed for identifying two laboratory parameters predictive of therapeutic success: erythrocyte counts and serum phosphorus dosage. Anemia is commonly observed in human and canine leishmaniasis and can originate from reduced production and/or increased destruction of erythrocytes [27,40,41]. During chronic inflammatory diseases, anemia results from low iron circulating levels, while inflammatory cytokines such TFN-α, IL-1 and IFN-γ inhibit erythropoiesis, and erythrocyte membrane damage by oxidizing agents shortens the lifespan of erythrocytes [42]. In chronic infections such as CanL, these damaged erythrocytes are retained and destroyed in the spleen, which becomes enlarged as a clinical sign of the disease. This is accompanied by a dysregulation of iron transporting proteins leading to the accumulation of storage iron [43,44]. This explains the normocytic normochromic type of anemia most dogs had in the beginning of the present study, which also demonstrates the improper iron supplementation during the treatment, as already described in the literature [45,46,47]. In chronic renal disease, which is common in CanL, anemia also results from decreased erythropoietin production in the kidney [8,48]. Kidneys are the main route for phosphorus excretion and hyperphosphatemia promotes progressive renal lesions [49], as described in dogs with late-stage CanL [50]. Serum globulin and acute phase proteins levels, proteinuria and UPC ratio are considered biomarkers for the clinical monitoring of dogs during leishmanicidal treatment and post-treatment [3,17,18,51,52]. However, very few studies have approached the identification of factors, to be used as reliable prognosis predictors, based on the outcome (death or cure) upon therapy [53]. In the present study, moderate-stage CanL dogs exhibited overall serum phosphorus—a marker of renal failure—within the reference range, but the logistic regression models clearly indicated that phosphorus’ in addition to erythrocyte’s changes were sensitive enough to predict chemotherapy success or failure.

Phosphorus interplays with calcium via modulation of several hormones and their serum concentration is approximately inversely related [50]. Overall, non-responder dogs of our study displayed lower magnesium, higher phosphorus and lower Ca/P ratio compared to responder dogs, with renal consequences demonstrated by elevated UPC ratio. Renal impairment-related ion imbalances underlie several clinical manifestations; for instance, magnesium is an enzymatic cofactor that participates in diverse metabolic reactions [40,54], including those related to erythropoiesis [55,56].

Cholinesterase and albumin serum levels normalized in responder dogs after six months of therapy, unlike in non-responders. As the liver is a site of *L. infantum* activity [57], this discrete interference with hepatic synthetic function could explain our findings in non-responder dogs. Here, serum urea and creatinine were within the reference values in both group of dogs. This corroborates literature data showing that azotemia is an uncommon finding, despite elevated frequencies of renal pathologies in CanL [58]. Creatinine is a by-product of CK, whose levels are reduced in dogs with LCan [59]; therefore, we suggest that this bias of azotemia diagnosis in advanced CanL results from sub-concentration of creatinine and might lead to kidney disease underdiagnoses. It has been stressed in the literature that the active infection by *L. infantum* and CanL progression relates with immune complex-mediated disease [60], which plays an important role in the pathophysiology of diverse clinical manifestations, including immune-mediated hemolytic anemia, glomerular lesions and renal failure [38,61]. In this sense, prednisolone was chosen as an anti-inflammatory drug in the present study during the short-term in all dogs to minimize the formation and circulation of soluble immune complexes and reduce inflammation through an inhibitory effect on complement activation [62,63]. Glucocorticoid therapy benefits in the treatment of CanL have been described previously [20,25,26]. It is worth mentioning that several studies have shown that the short- or long-term oral glucocorticoid administration in dogs do not induce significant changes in the several hematological and biochemical parameters [64,65,66,67]. The four-week course of prednisolone did not seem to have interfered with the biomarkers evaluated herein, as evaluations at T1 did not behave differently than those seen at subsequent follow-up points, as demonstrated in Figure 2, Figure 3 and Figure 4. Upon corticotherapy at anti-inflammatory doses, a discrete elevation in liver enzymes is expected in dogs, as previously described [66,68].

Altogether, we developed a mathematical formula using erythrocytes and phosphorus as markers for a prognosis of antiparasitic therapy success in CanL at the time of first diagnosis. Nevertheless, the study displays the reduced cohort studied, and the use of second line anti-parasite drugs, given the constraints for therapeutic approaches to CanL in Brazil, as major limitations. Because this was a long-term clinical study, it resulted in a small number of dogs that were followed up until the last evaluation. Nevertheless, the gathered data was so consistent that it allowed the creation of the formula, which can be an initial guide for further studies with larger sample sizes and different methods of cross-validation and in cohorts treated with first-line drugs. Taken together, these results are highly encouraging and may represent a new paradigm for clinical assistance, allowing a precocious therapeutic interference based on an improved diagnosis. 

## Figures and Tables

**Figure 1 microorganisms-08-00745-f001:**
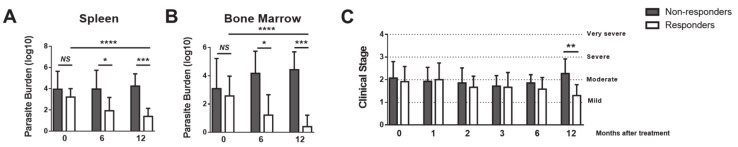
Responder dogs exhibited decreased parasite loads and improved the clinical staging. The canine cohort was separated into responder and non-responder dogs regarding the effect of anti-*Leishmania* chemotherapy. The parasite burden was quantified by qPCR in the spleen (**A**) and bone marrow (**B**) before and at 6- and 12-months post-treatment. The clinical stage of each dog was recorded at the time of the diagnosis (T0) and upon 1 (T1), 2 (T2), 3 (T3), 6 (T6) and 12 (T12) months of treatment (**C**). Data are shown as mean ± standard deviation (SD), Responder dogs (*n* = 12) Non responder dogs (*n* = 14). **p* < 0.05; ***p* < 0.01; ****p* < 0.001; *****p* < 0.0001.

**Figure 2 microorganisms-08-00745-f002:**
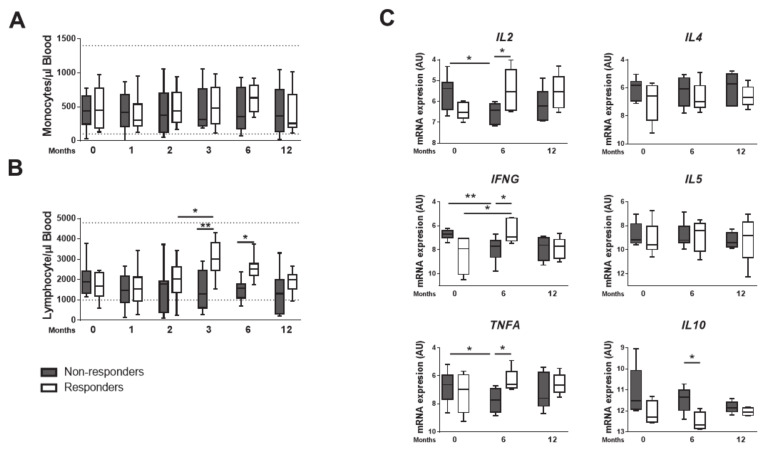
Responder dogs display Th1 signature. The serum absolute numbers of monocytes (**A**) and lymphocytes (**B**) was quantified from T0 until 12 months post treatment. mRNA was isolated from spleen biopsies at T0, T6 and T12. Quantitative PCR was performed for *IFNG*, *TNFA*, *IL2*, *IL4*, *IL5* and *IL10* (**C**). Data are shown as mean ± SD or in a box and whisker plot format, Responder dogs (*n* = 9) Non responder dogs (*n* = 8) * *p* < 0.05; ** *p* < 0.01.

**Figure 3 microorganisms-08-00745-f003:**
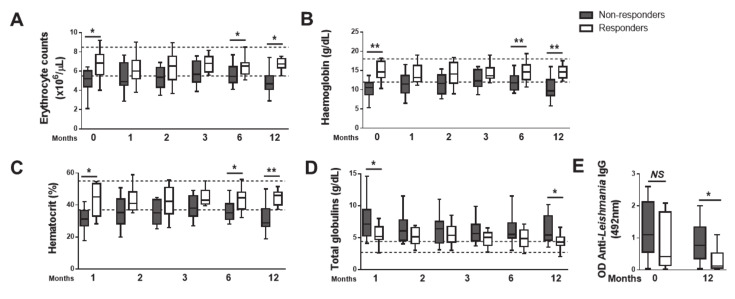
Success of anti-*Leishmania* therapy is associated with a restoration of erythrocyte, hemoglobin, hematocrit, as well as total and *Leishmania*-specific globulin values. Erythrocyte number (**A**) and the values of hemoglobin (**B**), hematocrit (**C**), total globulins (**D**) and anti-*Leishmania* IgG (**E**) were quantified in the serum at diagnosis (0) and upon 1, 2, 3, 6 and 12 months of anti-*Leishmania* treatment. Data are shown as box and whisker plot format, Responder dogs (*n* = 12), Non responder dogs (*n* = 14). * *p* < 0.05; ** *p* < 0.01.

**Figure 4 microorganisms-08-00745-f004:**
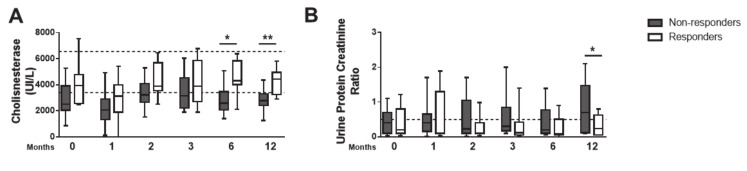
Non-responder dogs develop sub-clinical hepatic and renal lesions. Serum values of cholinesterase (**A**) and urine protein/creatinine ratio (**B**) were quantified at T0, T1, T2, T3, T6 and T12. Data are shown as mean ± SD or in a box and whisker plot format, Responder dogs (*n* = 12), Non responder dogs (*n* = 14). * *p* < 0.05; ** *p* < 0.01.

**Figure 5 microorganisms-08-00745-f005:**
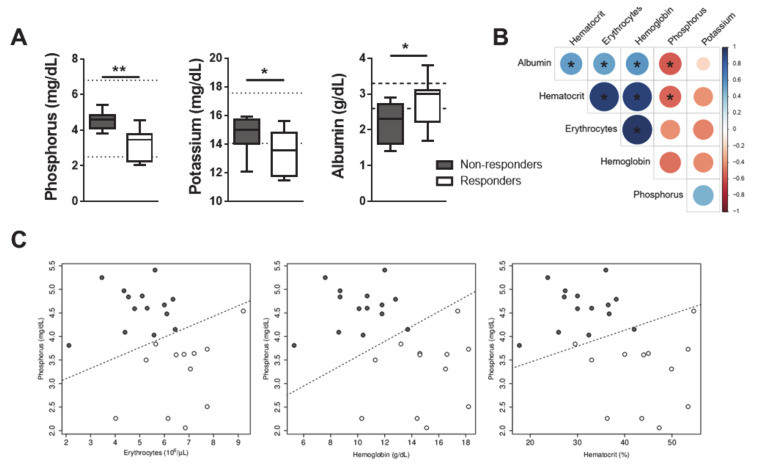
Development of a mathematical model with predictive value for the success of CanL chemotherapy including the measurement of the erythrocyte and phosphorus parameters. (**A**) The serum values of phosphorous, potassium and albumin were quantified at T0). (**B**) The Spearman linear correlation coefficient is shown for serum albumin, hematocrit, erythrocytes, hemoglobin, phosphorous and potassium values. (**C**) Scatterplots between phosphorous and hematocrit or erythrocytes or hemoglobin serum values. Data are shown as mean ± SD or in a box and whisker plot format. Responder dogs (*n* = 12), Non responder dogs (*n* = 14). **p* < 0.05; ***p* < 0.01.

**Table 1 microorganisms-08-00745-t001:** Values of hematological, serum biochemistry, oxidative stress parameters and parasitic load of the initial and final times of the responder and non-responder dogs to the treatment

Parameters	Responders (R)		*p-*Value (T0 vs T12)	Non Responders (NR)		*p-*Value (T0 vs. T12)	Normality Value	*p-*Value (T0 R vs. T0 NR)
Hematology	T0	T12		T0	T12			
Erythrocyte (x10^6^/μL)	6.68	6.73	0.497	5.01	4.97	0.940	5.5–8.5	0.033
Hemoglobin (g/dL)	15.03	14.75	0.771	10.21	10.54	0.885	12–18	0.0002
Hematocrit (%)	41.82	45.12	0.689	31.2	32.78	0.641	37–55	0.0023
Leucocytes (x10^3^/μL)	9.6	13	0.120	8.9	8,8	0.935	6–17	0.345
Lymphocytes (x10^3^/mm^3^)	1.69	1.92	0.373	1.92	1.41	0.181	1–4.8	0.406
Monocytes (/mm^3^)	493	424	0.606	457	429	0.972	150–1350	0.776
Reticulocytes (x10^3^/mm^3^)	90	34.6	0.144	51.6	46.1	0.678	<60	0.303
Serum biochemistry								
Total proteins (g/dL)	8.6	7.7	0.102	9	8.6	0.254	5.4–7.1	0.426
Albumin (g/dL)	2.79	3.04	0.167	2.21	2.54	0.260	2.6–3.3	0.017
Globulins (g/dL)	5.7	4.3	0.032	6.7	6.2	0.186	2.7–4.4	0.045
Albumin/Globulin (A/G) ratio	0,5	0,8	0.0682	0,4	0,5	0.242	0.5–1.7	0.081
Calcium (mg/dL)	11.73	10.50	0.330	10.87	10.39	0.218	9–11.3	0.064
GGT (U/L)	5.78	4.54	0.329	4.73	6.11	0.218	1–10	0.736
Creatinine (mg/dL)	1.26	1.0	0.082	1.0	1.2	0.330	0.5–1.4	0.920
Urea (mg/dL)	43.8	37.6	0.579	36.2	22.55	0.060	21–60	0.591
Cholesterol (mg/dL)	22.9	33.5	<0.0001	20.71	28.8	0.001	31–71	0.112
Magnesium (mg/dL)	1.62	2.22	0.264	1.66	1.86	0.017	1.8–2.4	0.104
HDL (mg/dL)	57.6	145.2	<0.0001	45.5	112.9	<0.0001	33–120	0.062
Triglycerides (mg/dL)	75.72	73.42	0.853	72.7	100	0.029	20–112	0.333
ALT (UI/L)	51.2	66.9	0.830	42.4	66.8	0.117	21–102	0.214
FA (UI/L)	74.97	86.82	0.587	101	124.7	0.933	20–156	0.255
AST (UI/L)	23.96	32.82	0.087	40.93	32.82	0.615	23–66	0.368
Glucose (mg/dL)	93.7	90	0.472	90.8	87.5	0.617	65–118	0.497
Iron (μg/dL)	280.07	85.29	0.094	360.4	142.2	0.075	30–180	0.149
Phosphorous (mg/dL)	3.24	2.84	0.401	4.61	3.074	0.011	2.5–6.0	0.0085
Potassium (mmol/L)	13.40	13.38	0.980	14.95	14.42	0.229	12.3–15.7 **	0.0023
CK (U/L)	47.08	152.39	<0.0001	37.35	87.79	0.106	100–200 **	0.504
CK-MB (U/L)	145.71	24.84	0.310	82.07	38.07	0.121	31–38.8	0.432
Cholinesterase (U/L)	4022.2	4200.5	0.737	3,198	2,807	0.911	1210–3020	0.057
UPC	0.48	0.31	0.113	0.44	0,85	0.342	0.5 ^#^	0.833
Serologic								
ELISA (OD)—Anti-*Leishmania* IgG	0.42	0.13	0.032	1.27	0.81	0.155	0.17**	0.324
**Parasite Load**								
Spleen (Parasite Burden (log10))	3.225	1.398	<0.0001	3.761	4.252	0.969		0.382
Bone marrow (Parasite Burden (log10))	2.580	0.408	<0.0001	2.879	2.579	0.431		0.748

Hematology and serum biochemistry normality value (NV) Source: Kaneko et al. [33]. NV—# UPC < 0.5 healthy animals; 0.5–1.0 bordering; >1.0 glomerular proteinuria. Source: Solano-Gallego et al. [5]. T0: time 0 (before treatment); T12: time 12 (12 months of treatment). ** NV based on the negative control values of the study. All significant values are shown in bold.

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
