# Peer review of "Mathematical Modelling Using Predictive Biomarkers for the Outcome of Canine Leishmaniasis upon Chemotherapy"

_microorganisms, 2020, doi:10.3390/microorganisms8050745_

Round 1

Reviewer 1 Report

The manuscript “Mathematical modelling using predictive biomarkers for the outcome of canine leishmaniasis upon chemotherapy “ gives very interesting information about the diagnosis and evaluation/follow up of dogs with leishmaniasis. However, the treatment for leishmaniasis used has not any evidence of efficacy, so the authors need to inform that it has been used just allopurinol as 3rd option of treatment of leishmaniasis.  Moreover, the authors used just allopurinol at dosage of 10 mg/kg/day. Furthermore the use of prednisone without any explanation could alters the results of all the parameters evaluated in this study. With this important concern, the authors need to rewrite all the manuscript, informing of the important limitations of the treatment and the use of steroids and, additionally, the discussion and conclusions need to be changed. Finally, there are several publications about prognosis or predictors factors of canine leishmaniasis that are not discussed in this manuscript. The manuscript at this moment needs to be rejected or needs major revisions before to evaluate its publication.  

Reviewer 2 Report

General comment

Authors present the results obtained in a cohort of naturally infected dogs (n=26) woth Leishmnia infantum (=L.chagasi) subjected to relatively non-conventional chemotherapy (metronidazole+ketoconazole+prednisolone for 40 days, followed by allopurinol treatment for at least 1 year). Animals were followed up by standard clinical biochemistry and hematology, immunology (specific antibodies, spleen cytokines), parasitology (parasite burden in spleen and bone marrow), clinics (classification of clinical status). Authors claim that an algorithm based on plasma phosphorous and erythrocytes number of the animal can be a predictor of the outcome of the treated dogs (responder/non responder). Canine leishmaniasis has a double interest since both humans and dogs are natural hosts and the latter are considered the main reservoir for human infections, and canine leishmaniasis is a first order pathology in veterinary clinics in many areas of the world. Thus, having prognostic markers to forecast the eventual clinical evolution of infected animals (treated or untreated) is clearly an important issue in the clinical management of the disease. The aim of the research is, therefore, interesting. There are, however, some issues that the authors must consider. Specific questions are listed below following the “natural” order and sections will be annotated when necessary.

Specific questions:

Introduction

l.51: irrelevant reference for the sentence. As a rule, references should be thoroughly revised: some are not adequate for purpose of the text; there are lacking references that could be better than the ones used (e.g. Sevá et al., 2016 PLoS ONE 11(7): e0160058. https://doi.org/10.1371/journal.pone.0160058; Lippi et al., 2014. JAVMA 245:1135-1140; Paltrinieri et al. 2016. Vet. Clin. Pathol. 45: 552-558; among other).

l.76. Reference (10), inadequate. Use one, of the many, reviews on chemotherapy (e.g. Oliva et al., 2010. JAVMA, 236: 36(11):1192-1198).

Material and methods

l.255-257. Please, explain. The study carried out (1 year) (l.266) was monitored by authors. The number of animals from the groups considered (R, NR) is not expressively given (figures or tables) (12-14?). Do this means that some animals were withdrawn from the study? When?

l.309-310. Leishmania sp share many antigens but if the authors are working with L.infantum, why L.donovani was used as antigen?

l.334-336: Table 1. That should be chnge to Supplementary material. There are not Materal and Methods but Results.

Table 1: In reviewer’s opinion two of the major indicators in the clinical flllow up of canine leishmaniasis are not given: Albumin/Globulin (A/G) ratio is standard in all areas endemic for leishmaniasis; non-regenerative anemia is a main indicator of clinical evolution and prognosis but it has not been included in the study. Why?

Phosphorous: total or ionized?

In general: what animals were classified at T0 as responder/non responder?. Alternatively, when the animals were classified (post hoc? At the end of the follow up? 12 months? 24 months?). Please, explain. If the classification was carried out with the outcome at time month 12, then there are some flaws: For instance, one of the parameters considered by the authors in their predictive algorithm , erythrocytes (along all the red cells parameters) was already below normality values in the first sampling: Doe this means that on T0 the animals were classified? Or it is a retrospective study (regression analysis) with all available numerical data (from the analyses) carried out at the end of the assay? This is most relevant, in reviewer’s view.

Results

l.94. Correct, according to the Fig. 1D, the survival % of non responder animals was 57%, not 67%. Incidentally: The animals were monitored for 1 year according to Material and Methods (at this time there was some casualty –fig. 1D: indicate). At this figure (1D) survival is indicated after 24 months. How can be included these data. More than probably the animals had very different clinical evolution (not monitored by the authors) and the possibility of additional infections with other agents cannot be ruled out. In any case, these results must not be included in the present paper.

Fig. 1: Although not significant the animals classified as NR had higher values of parasite burden in both spleen and bone marrow.

l.104: No whisker plot format here. Delete.

l.104: How many animals? 12-14?

Fig.2. Again, the values of NR animals were always less favorable, although non-significantly different (Th1 and Th2-related cytokines) (as in Fig. 1).

l.141: There was a reduction of specific anti-Leishmania IgG levels, as observed on Fig. 3E. However this results does not correspond to that given in Table 1: Responder: 1.03 to 0.54; non responder: 1.24 to 0.31 OD. This means that the reduction of specific antibodies was higher in non responder dogs. Explain the inconsistency.

Fig. 3. There were animals, at the first sampling, with anemia. Was treated in any way? Supplementary iron? Other treatments? Please, explain.

Fig.4: Again, the a posteriori considered on responder already showed a certain hepatic impairment (Fig. 4A).

l.178-179: When was carried out the discrimination?

Discussion

l.221-222: Phosphorous is a marker of renal failure. Please, check it.

l.230: the responder/non responder classification was carried out at month 24? It is suggested by the sentence but authors don’t know what happened with animals beyond month 12th. No samplings shown.

Minor
The use of prednisolone (and other corticosteroids) is not particularly recommended in the treatment of canine leishmaniasis (Adamama-Moraitou et al.2005. The Canadian Journal of Veterinary Research69:287–292) or, at least, very carefully monitored.

Round 2

Reviewer 1 Report

The authors have amended correctly all the previous concerns and doubts.

Author Response

We thank the reviewer for the careful revision that helped us in improving the manuscript. 

Reviewer 2 Report

A file has been added.

Author Response

We thank the reviewer for the critical revision of this manuscript that helped us in substantially improving our manuscript. 

Regarding the reviewer's question on this second version, we would like to add the following:

2. " Reviewer understands the interest from the authors (previous contribution of the group and limitations of approved drugs for canine leishmaniasis in Brazil) in the inclusion of the referenc by Nery et al., 2017. However, despite the surely relevant information, the journal where the research was published is not widely known. Translation of title and the name of the journal ( I guess it is not possible) does not help to readers not knowing Portuguese to read the contents. If the authors consider that it should be included (I don't) referencethey could be quoted alon the other references corresponding to second/third line drugs against leishmaniasis.

Authors reply: We have removed the reference Nery er al, 2017 from this sentence (line 79). The final sentence reads "This study was a retrospective longitudinal analysis of parasitological, biochemical and hematological parameters in a cohort of dogs treated for CanL with second-line antileishmanial drugs as reported in the literature." 

Ref.9-11: l. 61 & 62: Absence of parasitological cure in dog leishmaniasis is a well established fact (in any good review on chemotherapy of canine leishmaniasis). No need to include references about new areas (ref. 9) or particular trials with standard chemotherapeutic agents (ref. 10 & 11).

Authors reply: We followed the reviewer's suggestion. We have removed former references 9-11. The following references are now present.

Solano-Gallego, L.; Miró, G.; Koutinas, A.; Cardoso, L.; Pennisi, M.G.; Ferrer, L.; Bourdeau, P.; Oliva, G.; Baneth, G.; The LeishVet Group LeishVet guidelines for the practical management of canine leishmaniosis. Parasit. Vectors 2011, 86, 1-16.

Oliva, G.; Roura, X.; Crotti, A.; Maroli, M.; Castagnaro, M.; Gradoni, L.; Lubas, G.; Paltrinieri, S.; Zatelli, A.; Zini, E. Guidelines for treatment of leishmaniasis in dogs. J. Am. Vet. Med. Assoc. 2010, 236, 1192–1198.

L.71-72: Rephrase the sentence. In addition, ref. 17 refers only to a retrospective study in a non-endemic area (with infected dogs imported ) ; and no relationship was found between prognosis and specific antibody levels (!).

Authors reply: We have removed ref. 17 and rephrase the sentence to "Prognosis of response to treatment largely depends on the severity of the visceral disease, as determined by the clinicopathological alterations particularly those related to renal function [17,18]." We hope that the message is now clear.

As citations we used:

Roura, X.; Fondati, A.; Lubas, G.; Gradoni, L.; Maroli, M.; Oliva, G.; Paltrinieri, S.; Zatelli, A.; Zini, E. Prognosis and monitoring of leishmaniasis in dogs: A working group report. Vet. J. 2013, 198, 43–47

Maia, C.; Campino, L. Biomarkers Associated with Leishmania infantum Exposure, Infection, and Disease in Dogs. Front. Cell. Infect. Microbiol. 2018, 8, 1–18.

The reviewer also questioned in two distinct points (9 and 12) data that can be a result of the lower number of animals. We acknowledge this limitation and have rephrased a sentence in the discussion (line 275-281) to address this issue.

Round 3

Reviewer 2 Report

-